

# COVID-19: exploring impacts of the pandemic and lockdown on mental health of Pakistani students

Gul Muhammad Baloch[1], Sheela Sundarasen[2], Karuthan Chinna[1], Mohammad Nurunnabi[2], Kamilah Kamaludin[2], Heba Bakr Khoshaim[3], Syed Far Abid Hossain[4] and Areej AlSukayt[2]

[1] School of Medicine, Faculty of Medical and Health Sciences, Taylor's University, Subang Jaya, Selangor, Malaysia
[2] Department of Accounting, Prince Sultan University, Riyadh, Saudi Arabia
[3] Deanship of Admission and Registration, Prince Sultan University, Riyadh, Saudi Arabia
[4] College of Business Administration, International University of Business Agriculture and Technology, Dhaka, Bangladesh

Corresponding author
Gul Muhammad Baloch,
drgulbuledi@gmail.com

## ABSTRACT

**Background:** As of the present, the twenty-first century is experiencing what may be one of its most devastating events, in respect to infected and dead people by the virus. Now known to the world as COVID-19, the devastating disease of what has become a pandemic started its spread from Wuhan, China and swiftly engulfed the whole world with almost 11 million cases, in a span of around six months. It has not only increased the global burden of disease but has heavily dented many social institutions, including education.

**Methods:** This study investigates how the COVID-19 pandemic and subsequent measures of lockdown, quarantine, and social distancing have affected students. We look specifically into the effects on individuals' mental health, that is, the stress and anxiety levels of college and university students using the Zung Self-rating Anxiety Scale (SAS).

**Results:** Among 494 respondents, 61% were females, and the majority (77.3%) of the students were in the age group of 19–25 years. Among the respondents, 125 (25.3%), 45 (9.1%) and 34 (6.9%) experienced minimal to moderate, severe, and most extreme levels of anxiety, respectively. The variables of gender, age and year of study were significant at the 0.25 level by univariate analyses. Nevertheless, the ordinal regression indicates that only gender was significant. The odds of a female student being more anxious are higher compared to a male student (OR = 1.779, 95% CI [1.202–2.634], $P = 0.004$). The most prominent stressors attained from the qualitative feedback from the Pakistani students are associated with online teaching, concerns about their academic performance and completion of the current semester, uncertainty related to exam dates, and the status of the following semester.

**Conclusions:** This study will add to the existing body of literature on the impacts of the COVID-19 pandemic on the social and psychological health of students. The study outcomes will provide basic data for further applied and action research and a framework for universities and policy makers in Pakistan and the neighboring countries in the region with the same cultural contexts. Thus, relevant health interventions can be designed for better mental health and educational attainments of students from higher educational institutions. This pathological pandemic may

well lead to another pandemic of mental and behavioral illness. All stakeholders should join force regardless of pre-existing differences and inequalities to ensure the well-being of future generations, specifically students from higher educational institutions. The long-lasting impacts and the aftermath of this pandemic will unquestionably need further and future investigations. Keywords: Anxiety, students, mental health, COVID-19, Pakistan

## INTRODUCTION

### The beginnings

From the Paleolithic to Neolithic eras to this global village world, mobility has been one of the key characteristics of human civilizations. Humankind has been on the move not only with their material and non-material cultures but with their conditions of health and disease. Life lives together with death. According to the World Health Organization (WHO), "globally, about one billion cases of illness and millions of deaths occur every year from zoonoses" (*World Health Organization, 2020a*). Zoonoses are diseases transmitted to humans through animals. *Jones et al. (2008)* suggest that "with their peak incidence in the 1980s, the majority of emerging infectious diseases (71.8%) originate in wildlife and are increasing significantly over time." It is estimated that "more than six out of 10 known infectious diseases in people can be spread from animals and three out of every four new or emerging infectious diseases in people come from animals" (*Centers for Disease Control and Prevention, 2002*).

These emerging infectious diseases are not only the causes of alarming indicators of morbidity and mortality regarding the health of world populations but are a significant burden on global economies and the social fabric. As of today, the twenty-first century is experiencing what may be one of its most devastating events, second only in number of victims to the horrific toll of World War II. Now known to the world as COVID-19, the devastating disease started its spread from Wuhan, China in December 2019 and swiftly engulfed the whole world with more than eight million cases in a span of just around six months since the first of its reported cases (*World Health Organization, 2020b*). The mode of transmission of COVID-19 is through droplet infection.

### Current scenario: globally and pakistan

As of 12 July 2020, the number of confirmed cases peaked at 12,731,015 and caused 565,376 deaths over a span of six months since the first reported cases (*World Health Organization, 2020b*). This coronavirus was not new to the world, but is a causative microbe of severe acute respiratory syndrome (SARS), a form of animal coronaviruses that was documented in southern China in 2002–2003; it affected 29 countries of North and South America, Asia and Europe resulting in 8,098 cases and 774 fatalities (*Kahn & McIntosh, 2005*). On 31 December 2019, China first reported to the WHO Country Office in China the appearance of a type of pneumonia of unknown cause, in the city of Wuhan,

the capital city of Hubei Province. This outbreak of an epidemic spread rapidly from China to at least 20 other countries; thus, the Director General of WHO, having the authority in such a situation, on 30 January 2020 declared a Public Health Emergency of International Concern. A PHIEC is declared in accordance with the regulations of IHR 2005 (https://www.who.int/ihr/publications/9789241580496/en/). It was only on 11 February 2020 that WHO coined the name COVID-19 for this pandemic.

The first case of COVID-19 in Pakistan was detected on 25 February, and the first death reported on 29 March 2020 (*Nafees & Khan, 2020*). By the 67th report of the WHO on 29 March, Pakistan identified 14 deaths and 1,597 cases. The main sources of COVID-19 were individuals who were returning after visits for religious purposes to the neighboring countries of Iran (*Zaireen*), Saudi Arabia (pilgrims) and Malaysia (*Tablighi*), with Pakistani students from China. The *Tablighi* gathering of 14,500 participants in Malaysia in March 2020 had drawn about 1,400 foreigners, including from Pakistan (*Arumugam, 2020*).

The trajectory of COVID-19 infections in Pakistan shows only one confirmed case on 26 February, with 20 cases in two weeks' time, and the sharp epidemiological curve thereafter indicating a surge of cases that crossed 56,000 in two months on 25 May (*Worldometers, 2020*). This number is more than tripled in less than one month. As of 11 July 2020, the current number of confirmed cases in Pakistan was 246,351, with deaths at 5,123. (*World Health Organization, 2020c*).

The COVID-19 pandemic and the measures of lockdown and quarantine have created stress and anxiety, along with other predicaments for the general population, including students. When we are confronting a significant life change or traumatic event, we feel stressed. Stress is a response to a demand that triggers biological (physiological, chemical, hormonal, and emotional) changes in our body and brain. The term was coined by Hans Selye, "father of stress research", in 1936, describing it as "nonspecific response of the body to any demand" (*Tan & Yip, 2018*). Anxiety can become a mental health problem when it starts to impact one's ability to live one's day-to-day life smoothly. Anxiety is defined by the American Psychological Association (APA) as "an emotion characterized by feelings of tension, worried thoughts and physical changes" (*American Psychological Association, 2020*). Lockdown has been used by different countries globally as one of the strategies to contain the spread of COVID-19 in communities. Quarantine "separates and restricts the movement of people who were exposed to a contagious disease to see if they become sick" (*Centers for Disease Control and Prevention, 2000*). According to *Manuell & Cukor (2011)*, quarantine is "a key part of the overall public health strategy utilized during a pandemic". The suggested quarantine period is 2–14 days (*Fielding et al., 2015*; *Worldometers, 2020*). *Brooks et al. (2020)*, reported that the psychological impact of quarantine included "post-traumatic stress symptoms, irritability, insomnia, poor concentration, confusion, depression, low mood, emotional exhaustion and anger".

## Aim of the study

This study investigates how the COVID-19 pandemic and the subsequent measures of lockdown, quarantine, and social distancing have affected students. We are looking into

the effects and implications of all these on mental health; specifically, the stress and anxiety levels of college and university students. The sample population of this study comprised from all the major cities of Pakistan, including the districts form the urban and rural areas. The participant students belonged to Karachi, Hyderabad, Sukkur, Larkana, Lahore, Faisalabad, Rawalpindi and the capital city of Islamabad. This study will add to the existing body of literature on the impacts of the COVID-19 pandemic on the social and psychological health of students. The study outcomes will provide basic data for further applied and action research and the framework for universities and policy makers in Pakistan and the neighboring countries in the region with the same cultural contexts.

## LITERATURE REVIEW

### Global scenario: mental health impacts on the general population and students

There is not a single facet of life, whether it is sociological, psychological, or even economic, left untouched by the COVID-19 pandemic. *Rahman et al. (2020)*, in their systemic review, highlighted not only psychological effects but also the neurological effects of this pandemic. The neurological signs that they found in COVID-19 patients included "headache, nausea, vomiting, dizziness, loss of taste and smell and impaired consciousness". We can already observe the emergence of a kind of social stigma around COVID-19 in the form of xenophobic behaviors, posts and comments, albeit not everywhere. But undoubtedly this is the result of fear and anxiety in the population about the pandemic. In their study done on a cohort of SARS patients 30 months after the outbreak, *Mak et al. (2009)* concluded that "SARS can be regarded as a mental health catastrophe" that resulted in "PTSD and depressive disorders". *Haider, Tiwana & Tahir (2020)* are of the view that pandemics like COVID-19 increase stress levels and lead to psychiatric outcomes and that "it is possible that people may begin to experience transient mild to moderate depressive symptoms". It is not only that the COVID-19 patients are the only sufferers but "for other members of the public, without direct contact of the COVID-19 cases, mental health may be affected by preventive measures of social distancing, self-isolation and lockdowns" (*Haider, Tiwana & Tahir, 2020*).

Students, being part of the population and the institutions of society like family and education, are undoubtedly affected by epidemics. During the SARS outbreak in 2003, *Wong et al. (2004)* did a study with Hong Kong healthcare students, with non-healthcare students as the controls. Their "principle finding was that both healthcare and non-healthcare students were highly stressed" and there was no gender difference. The reasons behind the high levels of stress were different: nursing students were stressed due to a "perceived higher risk of infection due to prolonged contact with patients", while the non-healthcare students had "fear of the unknown". *Zeng, Jimba & Wakai (2005)* conducted a study on the psychosocial impact of SARS on students. However, the unique feature of their study was that they did the study on Chinese students who were living in Japan. They found 60% of the students had "fear, helplessness, worry and depression … even though none of them had SARS". Almost the same number of students (59.6%) "felt

an impact of SARS on their college life". Social discrimination was reported by 20% of respondents on the basis of ethnicity. A recent study by *Wang & Zhao (2020)* that explored the impacts of COVID-19 also found "higher anxiety" levels among university students. SAS was used in this study of 3,611 students, and the mean SAS score was 40.53, "significantly higher than the national norm of 29.78". Also, an important finding was that "female students showed more anxiety than male students". One of the main concerns of the students in this study was "the start of the new term", which would now be done online, instead of traditional face-to-face teaching.

### Pakistani scenario: mental health impacts on the general population and students

Mental health refers to the state of wellbeing in which individuals realize their own ability to cope with normal life stressors and productively work to contribute to their own community (*World Health Organization, 2005*). The COVID-19 pandemic, apart from the physical health-related signs and symptoms, poses serious threats to mental wellbeing and consequent changes in behaviors. A study done by *Balkhi et al. (2020)*, from the 400 respondents of the city of Karachi (where the very first case of COVID-19 in Pakistan was reported), shows that 62.5% of the respondents felt anxious on a daily basis, 88.8% feared going to market places, and 94.5% were concerned for the health of family members. The study also found "a higher tendency for graduates to fear for the safety of their health, even at home ($p < 0.01$)".

To stem the spread of COVID-19, along with other measures, on 13 March 2020, the Pakistan government announced the country-wide closure of all schools, colleges and universities (*Nafees & Khan, 2020*). *Aqeel et al. (2020)* suggest that this closure of educational students as a measure to contain the spread of COVID-19 also closed "a source of many students to cope with numerous personal and familial issues". This study found that students suffered anxiety and depression and that "young students who had exposure to the COVID-19 pandemic are more vulnerable to predisposition of mental health issues". In another study done with 347 university students from Pakistan by *Hongbo, Alishba & Muhammad (2020)* looking into predictors of anxiety, the authors found that students "appeared fearful of COVID-19, this fear was related to disgust sensitivity, anxiety sensitivity-related physical concerns, body vigilance, contamination cognitions and general distress". Yet another study done by *Salman et al. (2020)* with 1,134 students from higher education institutions of Pakistan found that COVID-19 has a "significant adverse impact on student's mental health" and "males had significantly less anxiety and depression scores than females"

## MATERIALS AND METHODS

### The study

Prior to the commencement of the study, ethical approval was obtained from the institutional review boards (IRB)[1] of the participating universities. Prince Sultan University Institutional Review Board (PSU IRB-2020-04-0038) and Research and Ethical Review Committee, Khairpur Medical College, Khairpur Mir's, Pakistan (KMC/

[1] IRB Prince Sultan University, Saudi Arabia (PSU IRB-2020-04-0038) and Research and Ethical Review Committee, Khairpur Medical College, Pakistan (KMC/RERC/29, Dated 30 May 2020).

RERC/29, Dated 30 May 2020) issued the ethical approval. A self-administered, validated questionnaire was developed to assess the stress and anxiety levels of university students in Pakistan during the smart lockdown period of COVID-19 from 26 May to 6 June 2020. The questionnaire was initially pilot tested. Subsequently, the final version of the questionnaire was sent via WhatsApp and emailed to university students using Google Forms. The questionnaire was unnamed, and informed consent was sought from students. The respondent students could only proceed to participate in the survey when they agreed and gave their consent after reading the aim of study and clicking 'yes' to the informed consent statement in the first part of the questionnaire. Participants were predominantly students from higher educational institutions residing in Pakistan. In determining sample size, we followed the guideline given by *Krejcie & Morgan (1970)*, which suggest for a large population a sample size of about 400 should be sufficient.

## The research instrument

In this survey, the students' anxiety levels were calculated using Zung's SAS, a validated 20-item self-report instrument. The instrument employs a Likert-type scale of 1–4 (1 = Never or very rarely; 2 = Sometimes; 3 = Often; 4 = Very often or always). Questions 1–5 characterized the emotional pointers of anxiety, whereas questions 6–20 inquired about physical anxiety symptoms. Emotional pointers of anxiety symptoms include depression, moodiness, irritability, feeling overwhelmed, loneliness and isolation while physical pointers of anxiety symptoms include restlessness, increased fatigability, difficulty in concentrating, irritability, muscle tension and sleep disturbance. As per design of the developer, for each respondent, the total anxiety score is obtained by adding the responses for the 20 items. The total score ranges from 20 to 80. The scores are then converted to an "Anxiety Index" with values ranging from 25 to 100. According to *Zung (1971)*, an Anxiety Index < 45 indicates "Anxiety within normal range", a value in the range of 45–59 indicates "Mild to moderate anxiety", a value in the range of 60–74 indicates "marked to severe anxiety", and values ≥ 75 indicates "Most extreme anxiety". In pilot test, the Cronbach's alpha value for internal consistency of the 20 items was 0.912.

## Data analysis

IBM SPSS version 22 (IBM SPSS Statistics for Windows, Version 22.0.: IBM Corp, Armonk, NY, USA) software was used in data analysis. Chi-square and ordinal logistic regression procedures were used to determine the factors associated with levels of anxiety. All the variables with p-values of less than 0.25 in the chi-square tests were tested in multivariate ordinal logistic regression analysis. For the final analysis, level of significance was set as 0.05

# RESULTS

## Demographics

A total of 494 completed questionnaires were received.

As shown in Table 1, among the 494 respondents, 61% were females and the majority (77.3%) of the students were in the age group of 19–25 years. More than one-third (39%) of the respondents were from the school of health sciences and almost the same

**Table 1 Demographic characteristics of the respondents.**

| Variable | Frequency | Percentage (%) |
|---|---|---|
| Gender | | |
| Female | 301 | 61.0 |
| Male | 193 | 39.0 |
| Age | | |
| Below 18 years | 45 | 9.1 |
| 19–25 years | 382 | 77.3 |
| Above 26 | 67 | 13.5 |
| Field of study | | |
| Engineering | 30 | 6.1 |
| Health Sciences | 193 | 39.1 |
| Management | 32 | 6.5 |
| Sciences | 98 | 19.8 |
| Social sciences | 141 | 28.5 |
| Level of Study | | |
| Undergraduate | 98 | 19.8 |
| Postgraduates | 26 | 5.3 |
| Professional | 370 | 74.9 |
| Year of study | | |
| Year 1 | 187 | 37.9 |
| Year 2 | 97 | 19.6 |
| Year 3 | 47 | 9.5 |
| Year 4 | 94 | 19.0 |
| Year 5 and above | 69 | 14.0 |
| Virtual learning | | |
| Yes | 341 | 69.0 |
| No | 111 | 22.5 |
| Not applicable | 42 | 8.5 |
| Current accommodation | | |
| College residency/hostel | 12 | 2.4 |
| Family Home | 449 | 90.9 |
| Rented place | 33 | 6.7 |

(38%) were in their first year of study. More than two-thirds of the students (69%) mentioned that their universities had started a virtual mode of delivery. In terms of accommodation, 91% were living in their family homes.

## Anxiety level

Among the respondents, 290 (58.7%), 125 (25.3%), 45 (9.1%) and 34 (6.9%) experienced minimal to moderate, marked to severe, and most extreme levels of anxiety, respectively. For further analysis, respondents in the marked to severe anxiety category and the most extreme anxiety category were grouped together as severe to extreme levels of anxiety. A summary of the results is shown in Table 2.

**Table 2  Anxiety level based on Zung's classification.**

| Anxiety | Frequency | Percentage | Anxiety | Frequency | Percentage |
|---|---|---|---|---|---|
| Normal | 290 | 58.7 | Normal | 290 | 58.7 |
| Mild–moderate | 125 | 25.3 | Minimal–moderate | 125 | 25.3 |
| marked–severe | 45 | 9.1 | Severe–extreme | 79 | 16.0 |
| Most extreme | 34 | 6.9 | | | |

**Table 3  Results from univariate analysis.**

| Variable | Normal (%) | Minimal–moderate (%) | Severe–extreme (%) | Chi-square | p-Value |
|---|---|---|---|---|---|
| Gender | | | | 15.010 | 0.001 |
| Female | 158(52.5) | 82(27.2) | 61(20.3) | | |
| Male | 132(68.4) | 43(22.3) | 18(9.3) | | |
| Age | | | | 6.331 | 0.176 |
| Below 18 years | 32(71.1) | 11(24.4) | 2(4.4) | | |
| 19–25 years | 216(56.5) | 99(25.9) | 67(17.5) | | |
| Above 26 | 42(62.7) | 15(22.4) | 10(14.9) | | |
| Field of study | | | | 5.781 | 0.672 |
| Engineering | 17(56.7) | 7(23.3) | 6(20.0) | | |
| Health Sciences | 118(61.1) | 49(25.4) | 26(13.5) | | |
| Management | 18(56.2) | 6(18.8) | 8(25.0) | | |
| Sciences | 59(60.2) | 27(27.6) | 12(12.2) | | |
| Social sciences | 78(55.3) | 36(25.5) | 27(19.1) | | |
| Level of Study | | | | 1.938 | 0.747 |
| Undergraduate | 221(59.7) | 99(21.1) | 60(16.2) | | |
| Postgraduates | 55(56.1) | 27(27.6) | 16(16.3) | | |
| Professional | 14(53.8) | 9(34.6) | 3(11.5) | | |
| Year of study | | | | 22.073 | 0.005 |
| Year 1 | 126(67.4) | 46(24.6) | 15(8.0) | | |
| Year 2 | 54(55.7) | 29(29.9) | 14(14.4) | | |
| Year 3 | 24(51.1) | 10(21.3) | 13(27.7) | | |
| Year 4 | 47(50.0) | 25(26.6) | 22(23.4) | | |
| Year 5 and above | 39(56.5) | 15(21.7) | 15(21.7) | | |
| Virtual learning | | | | 2.069 | 0.725 |
| Yes | 197(57.78) | 89(26.1) | 55(16.1) | | |
| No | 66(59.5) | 29(26.1) | 15(14.4) | | |
| Not applicable | 27(64.3) | 7(16.7) | 8(19.0) | | |
| Current accommodation | | | | 0.856 | 0.931 |
| College residency/hostel | 6(50.0) | 3(25.0) | 3(25.0) | | |
| Family Home | 265(59.0) | 113(25.2) | 71(15.8) | | |
| Rented place | 19(57.6) | 9(27.3) | 5(15.2) | | |
**Table 4 Results from ordinal multivariate analysis.**

| Parameter | B | SE | p-Value | OR_adj (95% CI) |
|---|---|---|---|---|
| Gender | | | | |
| Female | 0.576 | 0.200 | 0.004 | 1.779 [1.202–2.634] |
| Male | ref | | | 1 |
| Age | | | | |
| Below 18 years | −0.309 | 0.433 | 0.476 | 0.734 [0.314–1.715] |
| 19–25 years | 0.110 | 0.280 | 0.695 | 1.116 [0.645–1.933] |
| Above 26 | ref | | | 1 |
| Year of study | | | | |
| Year 1 | −0.372 | 0.297 | 0.211 | 0.689 [0.385–1.234] |
| Year 2 | 0.068 | 0.313 | 0.827 | 1.071 [0.579–1.979] |
| Year 3 | 0.293 | 0.372 | 0.431 | 1.341 [0.646–2.782] |
| Year 4 | 0.240 | 0.313 | 0.443 | 1.271 [0.688–2.349] |
| Year 5 and above | ref | | | 1 |

**Note:**

B, regression coefficient; SE, standard error; OR, odds ratio; CI, confidence interval.

For each respondent, the total anxiety score was obtained by adding the responses for the 20 items. The total score ranged from 20 to 80. The scores were then converted to an "Anxiety Index" with values ranging from 25 to 100. According to *Zung (1971)*, an Anxiety Index < 45 indicates "Anxiety within normal range", a value in the range of 45–59 indicates "Mild to moderate anxiety", a value in the range of 60–74 indicates "marked to severe anxiety", and values ≥ 75 indicates "Most extreme anxiety". Among the respondents in the sample, 125 (25.3%), 45 (9.1%) and 34 (6.9%) experienced minimal to moderate, marked to severe and most extreme levels of anxiety, respectively. For further analysis, cases with marked to severe anxiety and most extreme anxiety were grouped together and named as "severe to extreme" levels of anxiety.

## Factors associated with anxiety

### Univariate analysis

The results from Chi-squared analyses for the tests of associations between students' demographic variables and anxiety are presented in Table 3. Among the tested variables, only gender, age, and year of study were significant at the 0.25 level.

### Ordinal regression analysis

The variables of gender, age, and year of study that were significant at the 0.25 level in the univariate analyses were tested using ordinal multivariate analysis. Nevertheless, ordinal regression indicates that only gender was significant (Table 4). The odds of a female student being more anxious are higher compared to a male student (OR = 1.779, 95% CI [1.202–2.634], $P = 0.004$).

## DISCUSSION

The COVID-19 pandemic has been enormously stressful, and the student population has not been spared during this unprecedented period. Thus, this study explored the

mental health and especially the stress and anxiety levels of university students in Pakistan due to COVID-19, lockdown, and social distancing. The results indicate that approximately 41% experienced minimal to moderate, marked to severe, and most extreme levels of anxiety levels. Based on Zung's SAS classification, the percentage of anxiety levels in this study supports the findings of *Aqeel et al. (2020)*, who found "normal (43.2%), mild (20.5%), moderate (13.6%) and severe (22.7%) levels of anxiety prevalence" in their study of almost the same sample size ($N = 500$) of students from Pakistan. Similarly, *Salman et al. (2020)* found the proportions of students having "moderate-severe anxiety and depression (score = 10) were ~34% and 45%, respectively", which also supports the findings of this study.

Remarkably, the most prominent stressors attained from the qualitative feedback from the Pakistani students are associated with online teaching, concerns about their academic performance, and completion of the current semester, uncertainty related to exam dates, and the status of the following semester. This is also supported by a study by *Wang & Zhao (2020)*, whereby one of the major concerns of students is the start of the new term in which teaching would be online, instead of face-to face.

As practiced globally, to ensure continuity of classes, Pakistan universities have also moved to remote online classes almost overnight. The sudden move in the mode of teaching is on an unverified and unparalleled scale, thus causing massive implications from several perspectives. Universities have substituted face-to-face assessments with online assessments, again on an untested platform, thus creating anxiety and stress for the student fraternity. Most students, especially those living in villages and remote areas, have limited Internet access, and it added tremendous stress to these students. Economically, it was also not feasible for these students to spend money on Internet usage. It is equally distressing to note that some students had no laptops and had to borrow laptops and even cellphones from friends to participate in online classes. Students were also devastated by the high number of continuous assignments and assessments given by their lecturers. Many educators were equally unprepared by the sudden move to remote online classes and were thus executing their responsibilities in an impromptu manner, resulting in overwhelming assessments given to students, which seems to be the principal origin of stress for most students.

The results from multivariate ordinal regression analysis indicated that female students had a relatively higher level of anxiety compared to male students. This result supports the studies of *Azad et al. (2017)* and *Mirza & Jenkins (2004)*. Research indicates that females show greater anxiety than men and are comparatively more subject to anxiety disorders (*McLean & Anderson, 2009*). The authors further conjectured that the increased vulnerability of females to anxiety disorders can be comprehended by investigating sexual dimorphism in the underlying genetic factors that are known to have an impact on anxiety. A very recent study done by *Wang & Zhao (2020)* on the impacts of COVID-19 also found that females showed more anxiety than male students. A study by *Salman et al. (2020)* also supports the finding of this study: "males had significantly less anxiety and depression scores than females".

Additionally, most of the female students in this study stayed with their families due to the lockdown. Lockdowns and online classes are relatively new phenomena and parents, especially those from villages, did not fully comprehend the dilemma the students were enduring—having to cope with household commitments and care for their siblings in addition to attending online classes and dealing with overwhelming assignments and assessments. Thus, a lack of support from parents and family members has further contributed to the stress and anxiety levels of these university students in Pakistan. The evidence of greater impacts on females is supported by a study by *Hasan, Rehman & Zhang (2020)*, which found that "impacts of COVID-19 are likely compounded by the pre-existing inequalities in female students' access to education, especially at the secondary and tertiary levels".

These students were also socially distanced from their university mates, which is an imperative part of the young adult's life. Studies have shown a direct association between societal interaction and psychological wellbeing (*Dour et al., 2014*). Similarly, seclusion, lonesomeness, and communal remoteness impact the mental health of individuals (*Steptoe & Kivimäki, 2012*). Thus, the circumstances of the pandemic, lockdown and social distancing have impacted the psychological well-being of students, mainly female students.

## CONCLUSION AND FUTURE DIRECTIONS

This study explored the domain of psychological health (stress and anxiety) due to COVID-19 and the measures of lockdown and social distancing to contain the spread of the disease. Although both male and female students encountered stress and anxiety, females suffered higher levels of anxiety.

According to the *World Bank (2020)*, 63.334% of the Pakistani population lives in rural areas. With the sudden move to online teaching, the students faced difficulties due to the unaffordability and unavailability of laptops, unreliable Internet connections that affected their online teaching, assignments, exams and so on. The resultant scenario not only caused stress and anxiety related to academic calendars and careers but aggravated their mental health. *Hasan, Rehman & Zhang (2020)*, while investigating the possibilities of the students to study from home in Pakistan, found the "home-schooling challenges Pakistan's students face, given low rates of access to TV and the Internet".

Thus, it is equally important for the government and the education ministry to ensure the feasibility of the online teaching platform, as this study clearly indicates "online teaching" as a major stressor for students, specifically females. The government's position on mitigating the disruptive consequences of the COVID-19 pandemic on education is essential. The need for financial support at the state (federal and provincial) level for poor students of rural areas and expanded Internet access for both teaching and the academic performances of students should be prioritized. There is an urgent need for effective policies, considering the feedback from all stakeholders with regard to potential challenges faced by the education fraternity. Not only have this study and the COVID-19 pandemic highlighted the major constraints on transitioning the educational system but also exposed the prevalent patriarchy and its outcomes for females in Pakistan. Thus, due

consideration of female students' predicaments in this regard should be factored into all national education policies.

Higher educational institutions should also play a central role in identifying ways to mitigate students' mental health, stress, and anxiety levels. Universities should strive to develop and execute feasible action plans to alleviate students' mental health challenges, which are exacerbated by the COVID-19 pandemic and ensuing academic and career progression challenges. Universities should strategize innovative mechanisms and platforms for mental health counselling (i.e., tele-mental health counseling) and virtual meetings with students to constantly stay connected with students. Research indicates that tele-mental health is effective in treating anxiety and depressive symptoms (*Brenes et al., 2015*; *Dorsey & Topol, 2020*). Students should be encouraged to join online peer support groups to express common problems and obtain psychological support (*Rollman et al., 2018*), sharing coping strategies and so on, since lockdowns have removed an important and fundamental element of students' lives, that is, social support from friends.

In conclusion, we must recognize that COVID-19 has proven to be an extraordinary threat at the global level. We must understand that this pathological disease and pandemic may well lead to another pandemic of mental and behavioral illness. The long-lasting impacts and aftermath of this pandemic will unquestionably need further and future investigations.

## DRAWBACKS OF THE STUDY

This study, being a cross-sectional design, could not provide causal relationships. Thus, the future research studies should employ longitudinal design to capture the impending impacts on the mental health of students. This study used self-reported questionnaire and as with all self-reports, issues of subjectivity, reliability, and social desirability are unavoidable. Although the sample size of this study is not too small, but the sampling approach and the resultant participants may not reflect the bigger community of university students in Pakistan.

### Funding

This study was funded by the PSU COVID-19 Emergency Research Program (Grant ID: COVID19- CBA-2020-39). The funders had no role in study design, data collection and analysis, decision to publish, or preparation of the manuscript.

### Grant Disclosures

The following grant information was disclosed by the authors:
PSU COVID-19 Emergency Research Program: COVID19- CBA-2020-39.

### Competing Interests

The authors declare that they have no competing interests.

## Author Contributions

- Gul Muhammad Baloch conceived and designed the experiments, performed the experiments, analyzed the data, prepared figures and/or tables, authored or reviewed drafts of the paper, and approved the final draft.
- Sheela Sundarasen conceived and designed the experiments, performed the experiments, analyzed the data, prepared figures and/or tables, authored or reviewed drafts of the paper, and approved the final draft.
- Karuthan Chinna conceived and designed the experiments, performed the experiments, analyzed the data, prepared figures and/or tables, and approved the final draft.
- Mohammad Nurunnabi conceived and designed the experiments, prepared figures and/or tables, and approved the final draft.
- Kamilah Kamaludin analyzed the data, prepared figures and/or tables, and approved the final draft.
- Heba Bakr Khoshaim analyzed the data, prepared figures and/or tables, and approved the final draft.
- Syed Far Abid Hossain analyzed the data, prepared figures and/or tables, and approved the final draft.
- Areej AlSukayt analyzed the data, prepared figures and/or tables, and approved the final draft.

## Human Ethics

The following information was supplied relating to ethical approvals (i.e., approving body and any reference numbers):

Ethical approval was granted by Prince Sultan University Institutional Review Board (PSU IRB-2020-04-0038) and the Research and Ethical Review Committee, Khairpur Medical College, Pakistan (KMC/RERC/29).

## Data Availability

Raw data is available in the Supplemental Files.

## Supplemental Information

Supplemental information for this article can be found online at http://dx.doi.org/10.7717/peerj.10612#supplemental-information.

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
