# Peer review of "COVID-19: exploring impacts of the pandemic and lockdown on mental health of Pakistani students"

_PeerJ, doi:10.7717/peerj.10612_

## Round 0.1 · original submission · Major Revisions

Your manuscript is reviewed by three independent expert reviewers. All three reviewers have raised some important points. Reviewers have provided comments on methods, citations, and experimental design. I hope these comments will be helpful in improving the manuscript.

Reviewer 1 ·

Basic reporting

Though the article was written clearly and was lucid, there were instances where English could have been improved. However, instances of such mistakes were not too many. Consulting a language editor would really improve the quality of manuscript. Sufficient background information was provided. All the relevant references were mentioned. Article structure was fine. However, tables could have been presented in a more professional manner. They were too clumsy to look at. There is no need to display each and every grid of the table. The article was self-contained with relevant results to hypotheses.

Experimental design

Authors had well defined questions and pursued them rigorously. Methods were defined in sufficient details.

Validity of the findings

. It is well known that women are more prone to anxiety and depression compared to men. Given the poverty prevalent in Pakistan, not everyone can switch to online mode of education. Given the extreme patriarchal nature of Pakistani society, women, including students, do bear most of the brunt of household chores. This may reflect in poor academic performance. Hence, female students being significantly anxious does not surprise me.The findings of the study not very surprising.

Additional comments

It would have been great if authors mentioned that Covid-19 spreads through droplet infection, instead of giving analogy to a virus spreading through internet, and viruses not needing visa ( See lines 78-79).

Reviewer 2 ·

Basic reporting

SAS items tap both affective and somatic symptoms; there is no discussion of such symptoms.

Experimental design

Anxiety diagnosis is a clinical process and structured interviews need to be utilized apart from using scales. Moreover, recent research (Dunstan DA and Scott N, Depress Res Treat 2018:9250972, 2018) has questioned whether the existing cut-off for identifying the presence of a disorder might be lower than ideal. Mathematical formulas, including Youden’s Index and the Receiver Operating Characteristics Curve can be applied to positive diagnoses (presence of a disorder) from the clinical sample and negative diagnoses (absence of a disorder) from the community sample.

Validity of the findings

Findings need to be substantiated with more evidence/data by using multiple scales to analyze the effect confidently.

Additional comments

1) Section 1.3 is not needed in Introduction section.
2) Scientific deductions can only be made with valid citations. Citations in line 109, 126 do not seem appropriate.
3) Citations for many sections not included. (one example: line 170-173)\
4) Discussion is not cited at many places

Reviewer 3 ·

Basic reporting

Baloch et al. have studied the impact of the pandemic and lockdown amongst the Pakistani students. They have found that anxiety levels among women students are significantly more compared to male students. The manuscript is well written; however, it is still missing a few key points which are enlisted below. Therefore, at the current stage, this manuscript cannot be accepted, however, if authors can convince with a major revision and rewriting this manuscript explaining their results and subsequently providing their rationales, the decision can be reversed.

Experimental design

The authors should include the name of the cities and expand the area to comment on the whole country.

Research questions are very much important for the society and designing government plans to fight with the ongoing pandemic. However, the details of the statistical methods used in the study should be properly mentioned.

Validity of the findings

It lacks complete novelty, as because similar studies with COVID-19 and other pandemics are already available. However, earlier studies used a very limited sample size, so it would be better if the authors can increase the sample size significantly.

Additional comments

Major comments:
# No of samples (494) are severely low to comment on the anxiety level for the entire country. I recommend increasing the sample size significantly.
# Origin of the students and their locality (city wise) should be properly documented.
# It would be better to include students whose family or relatives are directly affected.
# previous studies, Azad et al. (2017) and Mirza & Jenkins (2004), had already shown similar findings with similar demographics, so proper justification and relevance of this study are completely lacking.
# Compare this study with previously reported the national anxiety level before this pandemic.
# Details of Statistical analyses are missing.
# I recommend including a section discussing the drawbacks of this study and future perspectives.

Minor comments:
# Data (the anxiety level among male and female students) should be shown as histograms and significance levels should be labeled and discussed properly in the text.
# Zung self-rating anxiety scale (SAS), the original paper should be properly cited.
# Azad et al. (2017) and Mirza & Jenkins (2004), both papers are missing in the reference section.
# Software used for Statistical analysis should be properly mentioned.

Annotated reviews are not available for download in order to protect the identity of reviewers who chose to remain anonymous.

---

## Round 0.2 · Minor Revisions

Reviewer 2 has some comments on the manuscript. Please remove the quotation below the title since it is not part of the title. These comments need to be addressed before the manuscript can be accepted for publication.

Reviewer 1 ·

Basic reporting

The manuscript reads much better now. The English language editing has made the manuscript easier to read.

Experimental design

After the revision, the experimental design is easy to understand.

Validity of the findings

The major revisions of manuscript have made the findings more valid.

Additional comments

I appreciate the efforts authors have put in to revise the manuscript thoroughly.

Reviewer 2 ·

Basic reporting

Raw data for Zung's classifications not shared.

Experimental design

As mentioned in the drawbacks, this study, being a cross-sectional design, could not provide causal relationships. The resultant participants may not reflect the bigger community of university students in Pakistan. The various data points of field of study and variate analyses does not provide any substantial result. Rigorous investigations does not seem to be performed.

Validity of the findings

No comment

Additional comments

Statements like: "Although both male and female students encountered stress and anxiety, females suffered higher levels of anxiety. This is likely the result of the patriarchal edifice of Pakistani society, where females are the perpetual sufferers of pre-existing inequalities" is not substantiated by results. The data is not robust

Reviewer 3 ·

Basic reporting

Now, the manuscript looks much better. Authors have successfully addressed all the major concerns.

Experimental design

All the relevant pieces of information regarding the statistics are now included.

Validity of the findings

No comments

Additional comments

No comments

---

## Round 0.3 · accepted · Accept

I would like to thank the authors for making the necessary changes. I recommend acceptance of this article.